# The Effect of the 2019 Novel Coronavirus Pandemic on Stroke and TIA Patient Admissions: Perspectives and Risk Factors

**DOI:** 10.3390/jcm10071357

**Published:** 2021-03-25

**Authors:** Luke Carson, Christopher Kui, Gemma Smith, Anand K. Dixit

**Affiliations:** 1Newcastle-Upon-Tyne Hospitals NHS Foundation Trust, Newcastle-Upon-Tyne NE1 4LP, UK; luke.carson@nhs.net (L.C.); christopher.kui@nhs.net (C.K.); 2Newcastle University Clinical Academic Office, Newcastle-Upon-Tyne NE1 7RU, UK; 3County Durham and Darlington NHS Foundation Trust, Durham DH1 5TW, UK; gemmamarie.smith@nhs.net

**Keywords:** stroke, TIA, coronavirus, COVID-19

## Abstract

Background: The 2019 novel coronavirus pandemic has generated concern from stroke specialist centres across the globe. Reductions in stroke admissions have been reported, despite many expecting an increase due to the pro-thrombotic nature of 2019 novel coronavirus. Aims: To assess the impact of the pandemic and subsequent lockdown on stroke admissions and transient ischaemic attack referrals at the Royal Victoria Infirmary, Newcastle-Upon-Tyne, and additionally on patient behaviours affecting modifiable risk factors or perspectives related to accessing healthcare. Methods: A single-centre retrospective data analysis was carried out on a “lockdown” cohort of suspected stroke patients admitted between 11 March to 26 May 2020 and a “pre-lockdown” cohort admitted in 2019. Differences in weekly admissions, weekly referrals, onset-to-presentation time and weekly thrombolysis cases were examined. Further analysis interrogated these cohorts separated by Bamford classification and stroke mimics (such as seizure/hemiplegic migraine/functional neurology). A binary-format questionnaire was separately administered to admitted patients from 15 April to 5 June 2020. Results: Significant reductions in weekly posterior circulation infarct (−43%, *p* = 0.017) and stroke-mimic (−47%, *p* < 0.001) admissions and weekly referrals diagnosed as non-transient ischaemic attack (−55%, *p* = 0.002) were observed in the lockdown cohort, with no differences in onset-to-presentation time. Over 25% of questionnaire respondents reported less physical activity, increased isolation and delaying their presentation due to the pandemic. Conclusions: This study provides evidence of reduced stroke-mimic and posterior circulation infarct admissions. Questionnaire findings suggest that patients need to be informed to ensure they appropriately seek medical advice. Significant communication at the stroke-primary care interface is needed to support referral pathways and management of modifiable risk factors.

## 1. Introduction

The 2019 novel coronavirus (COVID-19) outbreak has marked the transition into a new decade. Originating in Wuhan, China, the COVID-19 rapidly spread across the globe, consequently being declared a pandemic by the World Health Organisation (WHO) on 11 March 2020 [1,2]. As hospitals across the United Kingdom (UK) redeployed staff and increased bed capacity to prepare for a surge in hospitalisations, non-COVID-19-related admissions fell to an all-time low [3].

Early reports from Italy confirmed a similar trend for stroke; hospitals across the country noted an almost 50% reduction in stroke unit admissions compared to the previous year [4], despite COVID-19 being positively associated with hypercoagulability [5,6], increased risk of stroke and increased neurological severity of stroke [7,8,9]. These conflicting observations suggest the reduction in admissions may arise from altered patient behaviour rather than reduced stroke incidence during the COVID-19 pandemic, a concerning explanation given the significantly improved outcomes associated from early presentation to a specialist stroke unit [10,11].

Following the WHO statement declaring COVID-19 to be a pandemic and amidst rising incidence and mortality, the British government formally declared nationwide lockdown (which were later relaxed in July 2020). The UK government told its population to stay at home, leaving only to shop for basic necessities, for once daily exercise, for medical need or for key workers to go to work. Residents of the UK faced significantly restricted freedom of movement for the first time since World War II, potentially affecting modifiable risk factors for stroke/transient ischaemic attack (TIA) whilst introducing new psychosocial elements which may interfere with the timely presentation to a healthcare professional. Establishing how the COVID-19 pandemic has affected patient perspectives and behaviours could provide crucial guidance on how to manage stroke services during further waves of COVID-19 or in any future pandemic.

### Aims

This study aimed to assess whether the COVID-19 pandemic led to a reduction in stroke/TIA admissions at a single hyperacute stroke unit based in the Royal Victoria Infirmary, Newcastle-Upon-Tyne, UK. A further exploration was undertaken via questionnaire to identify whether the pandemic affected the perspectives and behaviours of stroke patients, contributing towards altered modifiable risk factors or delayed presentation to a specialist stroke unit and a primary stroke centre.

## 2. Materials and Methods

### 2.1. Specialist Stroke Unit

The Royal Victoria Infirmary (RVI) hyperacute stroke unit (HASU) is a specialist stroke unit which runs a 24/7 service providing consultant-led thrombolysis alongside a stroke specialist nurse. The HASU admits an annual average of 1250 patients from the cities of Newcastle-Upon-Tyne and Gateshead, UK; this includes referrals from neighbouring primary stroke centres requiring mechanical thrombectomy or decompressive hemicraniectomy.

A stroke specialist nurse is on-site 24/7, accepting paramedic calls and triaging patients directly to the HASU between 08:00–20:00 and via the emergency department (ED) overnight. Stroke physicians receive ambulance pre-alerts for potential thrombolysis candidates prior to the patient reaching the hospital and are actively engaged in triaging suspected stroke patients in the ED. Mechanical thrombectomy services are provided six-days a week from 09:00–16:00. Finally, patients referred for the seven-day TIA service are reviewed at outpatient clinic during weekdays and at the HASU on weekends. A formal internal organisation-wide audit revealed no differences in stroke services or HASU admission criteria between 2018 and 2020.

### 2.2. Admissions Data Analysis

A single-centre retrospective analysis was carried out on a stroke admission and TIA referral database maintained by stroke specialist nurses at the RVI HASU. The database is used to populate the Safe Implementation of Treatment in Stroke (SITS) and Sentinel Stroke National Audit Programme (SSNAP) registries. Patients in the “lockdown” cohort presented during the 11-week period between 11 March 2020 and 26 May 2020 and patients in the “pre-lockdown” cohort presented during the same-length period between 11 March 2019 and 26 May 2019. All patients admitted as a potential stroke were deemed relevant and therefore included in our analysis.

To measure the impact of COVID-19, the following comparisons were made between the two cohorts: (1) weekly number of stroke admissions, (2) weekly number of patients thrombolysed, (3) weekly number of TIA referrals, (4) stroke onset-to-admission time. Further analysis similarly interrogated those presenting with intracerebral haemorrhage (ICH), ischaemic strokes, ischaemic strokes separated by Bamford classification and stroke mimics (seizure/hemiplegic migraine/functional neurology or other pathology mimicking stroke such as a brain tumour). Statistical analyses were performed in RStudio using the “ggpubr” and “dplyr” packages [12,13,14,15]. Continuous variables were examined via Student’s unpaired t-test (if normally distributed as per the Shapiro–Wilk test) or Wilcoxon’s signed-rank test, whilst categorical variables were examined via Fisher’s Exact test.

### 2.3. COVID-19 Patient Questionnaire

A patient questionnaire was designed (Figure 1) and added to the routine stroke clerking process to gauge the impact of the COVID-19 pandemic and associated lockdown on patient perspectives and behaviours. The questions focused on identifying changes in modifiable risk factors or perceived barriers to a timely presentation to allow appropriate ACT-FAST guidance to be given on discharge. 

The questionnaire was administered at the RVI HASU from 15 April 2020 to 1 June 2020 and additionally at the University Hospital of North Durham (UHND) Stroke Unit (a neighbouring primary stroke centre) from 15 May 2020 to 5 June 2020. All patients who presented with suspected stroke were included in the analysis, as ACT-FAST guidance would be relevant. The questionnaire was omitted if patients were unable to communicate (with no household members available), approaching end-of-life, or taken straight to intensive care. 

Results were recorded as a binary yes/no format and reported as percentages. In addition to being retrospectively analysed for this study, the answers given were used to provide tailored advice on discharge (such as reassurance and discussions around secondary prevention and ACT-FAST guidance).

## 3. Results

### 3.1. Stroke Admissions

Between 11 March 2020 and 26 May 2020, 200 stroke admissions were registered (lockdown cohort); a 10% reduction when compared with 223 admissions during the same 11-week period in 2019 (pre-lockdown cohort) (Table 1). There was no significant difference (95% CI −2.01–6.19, *p* = 0.301) in weekly stroke admissions between the pre-lockdown (mean 20.3 ± SD 5.1) and lockdown (18.2 ± 4.1) cohorts, as was the case for weekly ICH (95% CI 1.73–2.36, *p* = 0.227), ischaemic stroke (95% CI −1.02–6.47, *p* = 0.144), total anterior circulation infarct (TACI) (95% CI −1.03–3.03, *p* = 0.316), lacunar infarct (LACI) (95% CI −1.38–1.92, *p* = 0.734) and partial anterior circulation infarct (PACI) (median 4, IQR 3.5 pre-lockdown vs. 4, 2.5 lockdown, *p* = 0.616) admissions (Table 1, Figure 2).

However, weekly posterior circulation infarct (POCI) admissions were significantly reduced (95% CI 0.328–2.94, *p* = 0.017) between the pre-lockdown (mean 3.82 ± SD 1.7) and lockdown (2.18 ± 1.2) cohorts, as were the weekly thrombolysed stroke admissions (median 4, IQR 1.5 pre-lockdown vs. median 3, IQR 2.0 lockdown, *p* = 0.018) (Table 1, Figure 2).

In terms of the stroke subtypes, the proportion of ischaemic stroke versus ICH admissions in the pre-lockdown (*n* = 204/223, 92% ischaemic stroke) and lockdown (*n* = 174/200, 87% ischaemic stroke) were similar (*p* = 0.156). Despite the reduction in mean weekly POCI admissions described above (Table 1), there were no differences between the two cohorts in terms of ischaemic stroke Bamford-classification subtypes (Table 2). Finally, the total proportion of ischaemic strokes thrombolysed was reduced by 7.7% between the pre-lockdown (*n* = 45/204, 22% thrombolysed) and lockdown (*n* = 25/174, 14% thrombolysed) cohorts, but this was non-significant (*p* = 0.063) (Table 2, Figure 2).

An accurate time (in hours) from symptom onset to hospital presentation was identified for 78% of the pre-lockdown (*n* = 175/223) and 83% of the lockdown (*n* = 165/200) cohorts. For all patients admitted with stroke, the median onset-to-presentation time was similar between both cohorts (median 4.83, IQR 13.1 pre-lockdown vs. median 5.42, IQR 13.1 lockdown, *p* = 0.323), as was the case for patients admitted with ischaemic stroke (*p* = 0.249), ICH (*p* = 0.673), TACI (*p* = 0.218), PACI (*p* = 0.222), LACI (*p* = 0.833) and POCI (*p* = 0.401) (Table 3). 

### 3.2. Stroke Mimic Admissions and TIA Referrals

There was a 47% reduction in total stroke mimic/non-stroke admissions between the pre-lockdown (*n* = 193) and lockdown (*n* = 102) cohorts. The weekly number of stroke mimic admissions was significantly reduced in the lockdown cohort (mean 17.5 ± SD 3.3 pre-lockdown vs. 9.30 ± 4.1 lockdown, 95% CI 4.05–11.6, *p* < 0.001) (Table 4) (Figure 3).

Similarly, there were 31% fewer referrals made to the RVI HASU in the lockdown cohort (*n* = 105) when compared to the pre-lockdown cohort (*n* = 152). This difference was reflected as a significant reduction in weekly referrals (median 12, IQR 6.5 pre-lockdown vs. median 9, IQR 5.5 lockdown, *p* = 0.037). Of note, the number of weekly referrals eventually diagnosed as TIA remained similar between both cohorts (*p* = 0.552), but there was a significant 45% reduction in weekly referrals diagnosed as non-TIA in the lockdown cohort (mean 8.64 ± SD 3.1 pre-lockdown vs. 4.73 ± 2.1 lockdown, 95% CI 4.73–8.64, *p* = 0.002) (Table 4) (Figure 3).

### 3.3. COVID-19 Patient Questionnaire

A total of 216 patients admitted with suspected stroke were screened for questionnaire suitability; 27% were excluded (*n* = 59/216) and the answers from 157 patients were successfully recorded (*n* = 93/157 RVI, *n* = 64/157 UHND) using a pro-forma (Figure 1).

In terms of modifiable risk factors: 35% of patients (*n* = 55/157) reported an adversely affected exercise/physical activity schedule due to the COVID-19 pandemic, with an unhealthier diet in 14% (*n* = 22/157), increased smoking in 7% (*n* = 11/157) and increased alcohol consumption in 9% (*n* = 14/157). 

Upon exploring perceptions and barriers towards accessing healthcare, 14% (*n* = 22/157) ignored symptoms and 25% (*n* = 39/157) delayed their current presentation (of suspected stroke) to hospital due to the pandemic. In addition, 14% (*n* = 22/157) missed routine general practitioner (GP) reviews, 5% (*n* = 7/157) struggled to collect prescriptions, 5% (*n* = 7/157) reported medication non-compliance and 25% (*n* = 39/157) had been alone in the house more often due the pandemic.

## 4. Discussion

In contrast to two large multicentre studies which reported reductions in ischaemic stroke and TIA [16,17] admissions, the RVI”s HASU experienced no significant reductions in weekly stroke, ischaemic stroke, or ICH admissions. A significant 31% fewer weekly referrals were made for suspected TIA, though the number of eventual TIA diagnoses remained similar. Frisullo et al. shared similar negative findings which may be reflective of a smaller cohort size [18]. Hoyer et al. reported that reductions in ischaemic stroke and TIA diagnoses were only seen in 2/4 and 3/4 centres recruited, highlighting that the impact of COVID-19 may depend on local factors such as catchment area [16].

Our study reports a significant 43% reduction in weekly POCI admissions, whilst the overall proportion of ischaemic stroke subtypes when separated by Bamford-classification was unchanged in the pre-lockdown and lockdown cohorts. Similarly, there was a highly significant 47% reduction in weekly stroke mimic/non-stroke admissions and a significant 55% reduction in weekly referrals made for TIA eventually diagnosed as non-stroke. POCIs lack representation in the ACT-FAST campaign and symptoms of dizziness or nausea enhance the likelihood of misdiagnosis in the emergency department [19], whilst stroke mimics most frequently present “mildly” with a NIHSS score of 0 [20]. During the pandemic, patients may have a higher threshold for seeking healthcare advice, with anecdotal evidence suggesting fears of virus transmission or burdening the health service [21,22]. Milder POCI and stroke mimic symptoms may not be concerning to a patient; certainly, 14% of questionnaire respondents in this study ignored symptoms that they would normally present with. Even if GP advice is sought, primary care teams themselves may have higher thresholds for referral to hospital services, especially for perceived low-risk patients.

A major concern regarding the impact of COVID-19 in acute stroke relates to delayed hospital presentation, which could place patients out of the thrombolysis time window (of 4.5 h); 25% of patients across the RVI and UHND reported delaying their presentation due to the ongoing pandemic. A further 25% reported being alone at home more often than usual, highlighting at-risk individuals who may not be able to seek help if acutely unwell or whose symptoms may go unnoticed for longer, as has been previously suggested [23]. Unexpectedly, the median time from symptom onset to hospital presentation was unchanged in the lockdown cohort of this study, across all stroke subtypes. Diegoli et al. reported similar findings, but noted a higher observed proportion of severe NIHSS strokes during the pandemic may have skewed towards faster presentations [17]. Reassuringly, while fewer stroke admissions were being thrombolysed each week, the total proportion of ischaemic strokes thrombolysed remained unchanged in this study, largely in line with other reports [16,17].

Finally, questionnaire results collected from the RVI and UHND have emphasised the impact of the COVID-19 pandemic and resulting lockdown on modifiable risk factors for stroke. In terms of lifestyle, admitted stroke patients reported reduced exercise routines (35%), unhealthier diets (14%), increased smoking (7%) and alcohol consumption (9%). Long-term prevention therapies for stroke may also be affected to a lesser extent, as some patients reported increased medication non-compliance (5%) and difficulty in collecting prescriptions (5%). The pandemic’s impact on primary care is yet to be fully quantified, with services understandably having shifted towards telephone consultations and suspension of review appointments. Due to the pandemic, around 14% of admitted patients reported missing GP reviews; in the context of stroke, such appointments are crucial for monitoring and managing risk factors such as hypertension, cholesterol and lifestyle habits.

Lockdown measures in the UK had been reduced then re-introduced due to repeated waves of COVID-19. The government and public’s attention are at the point of writing fully focussed on vaccination efforts and prevention of further waves of the pandemic. Internal audits within the RVI’s HASU revealed no changes in service provision or admission/referral criteria; the findings discussed point towards the importance of enhancing public awareness and managing perspectives to ensure patients appropriately seek medical advice with concerning symptoms, especially in time-sensitive stroke pathways and with nefarious stroke mimic diagnoses. In the further waves of this or any pandemic, stroke services will need to work closely with primary care teams to maintain referral pathways and ensure minor or transient features indicative of stroke or TIA are not being missed. Especially in the lockdown setting, small interventions such as self-monitoring of blood pressure and the guarantee of safe, home-assessment options will be crucial, highlighting the undeniable role of GPs in keeping patients informed, reassured and healthy.

## 5. Limitations

This study reported findings from the retrospective analysis of admissions and referrals data undertaken at a single HASU in the UK, with questionnaire data additionally obtained from a neighbouring primary stroke centre. An internal audit was undertaken which confirmed no changes in stroke services provision; however, the lack of patient-level data (for example NIHSS scores) could introduce biases and the small sample size may mask significant findings. The questionnaire results have no comparison population at present and due to the binary format can only offer an initial insight into patient factors at present.

## 6. Conclusions

This study presents an exploration of stroke patient behaviours during the COVID-19 pandemic and resultant lockdown and an insight into the negative impacts on modifiable risk factors and patient perspectives towards seeking medical help. In addition, it adds novel evidence of reduced POCI and stroke-mimic admissions whilst providing reassurance that most patients present in a timely fashion, appropriately receive time-sensitive therapies and can still access TIA referral pathways. To expand on these results, future multi-centre studies may incorporate primary care referral rates and consider subgroup analyses on Bamford-classification subtype and NIHSS severity, Structured interviews may further illuminate stroke service providers on the impact of COVID-19 on patients and guide public health strategies.

## Figures and Tables

**Figure 1 jcm-10-01357-f001:**
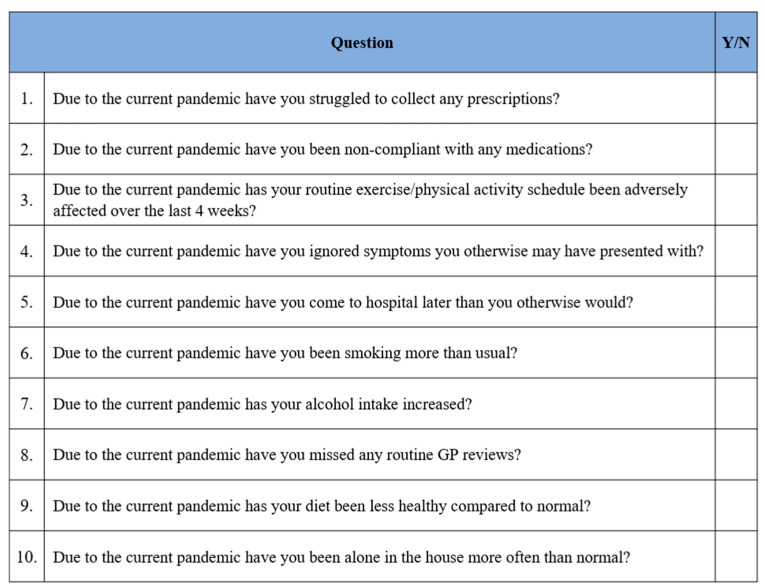
2019 novel Coronavirus Patient Questionnaire. The questions were designed to identify the impact of lockdown on modifiable risk factors (concordance, lifestyle) and timely presentation to a healthcare professional due to perceived barriers, with answers recorded in a Y/N (Yes/No) binary format.

**Figure 2 jcm-10-01357-f002:**
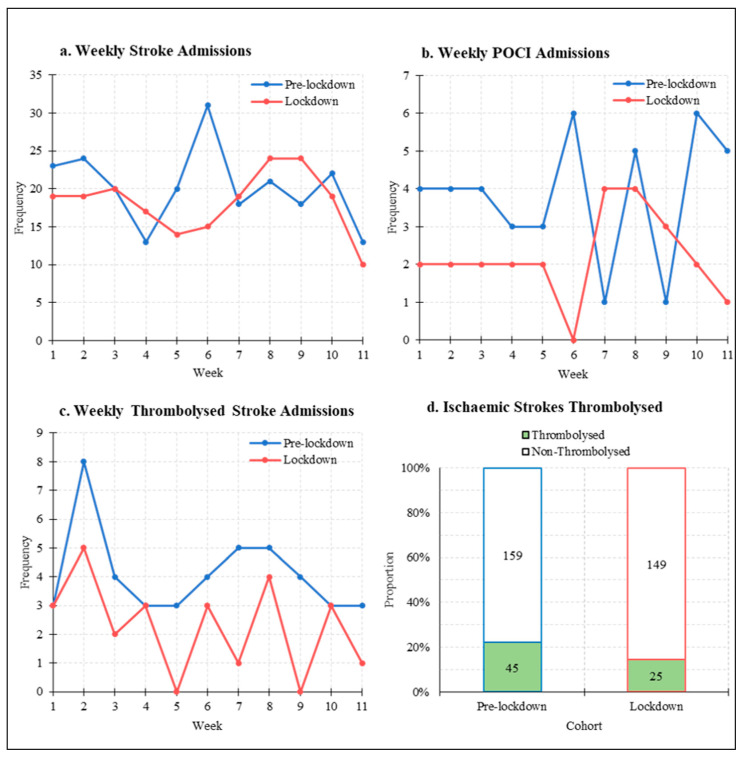
Stroke Admissions Trends in the Pre-lockdown and Lockdown Cohorts. Weekly (**a**) stroke, (**b**) POCI, (**c**) thrombolysed stroke admissions data in the pre-lockdown (blue) and lockdown (red) cohorts. Additional analysis of ischaemic stroke admissions (**d**) revealed no difference in the proportion of ischaemic strokes thrombolysed. POCI: posterior circulation infarct.

**Figure 3 jcm-10-01357-f003:**
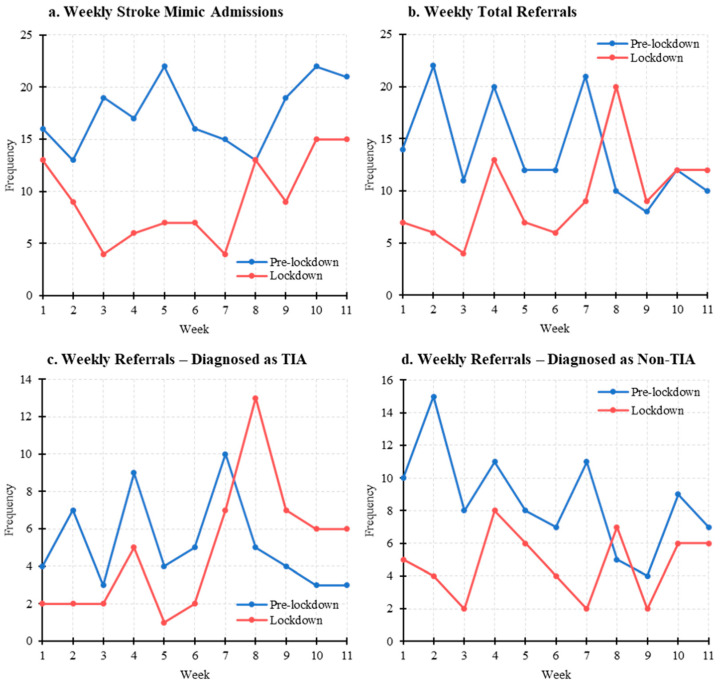
Stroke Mimic Admissions and TIA Referral Trends in the Pre-lockdown and Lockdown Cohorts. Weekly (**a**) stroke-mimic/non-stroke admissions, (**b**) total referrals made to the Royal Victoria Infirmary Hyperacute Stroke Unit, (**c**) referrals diagnosed as TIA, (**d**) referrals diagnosed as non-TIA data in the pre-lockdown (blue) and lockdown (red) cohorts. TIA: transient ischaemic attack.

**Table 1 jcm-10-01357-t001:** Stroke Admissions in the Pre-lockdown and Lockdown Cohorts. Stroke admissions data in the lockdown and pre-lockdown cohorts. There were significantly reduced weekly posterior circulation infarcts and thrombolysed stroke admissions in the lockdown cohort. *p* values were generated with Student’s t-test if data were normally distributed, otherwise Wilcoxon’s signed-rank test was performed; values <0.05 were deemed statistically significant. All stroke: All ischaemic and haemorrhagic strokes admitted, ICH: intracerebral haemorrhage, TACI: total anterior circulation infarct, PACI: partial anterior circulation infarct, LACI: lacunar infarct, POCI: posterior circulation infarct: NOS: not otherwise specified, SD: standard deviation, IQR: interquartile range, CI: confidence intervals, NA: not available.

	11-Week Total			Weekly Admissions			
Diagnosis	Pre- Lockdown	Lockdown	Change	Pre-Lockdown (Mean ± SD)	Lockdown (Mean ± SD)	95% CI	*p* Value
All stroke	*n* = 223	*n* = 200	−10.3%	20.3 ± 5.1	18.2 ± 4.1	−2.01–6.20	0.301
ICH	*n* = 19	*n* = 26	+26.9%	1.73 ± 0.9	2.36 ± 1.4	1.73–2.36	0.227
Ischaemic Stroke	*n* = 204	*n* = 174	−14.7%	18.5 ± 4.6	15.8 ± 3.8	−1.02–6.47	0.144
TACI	*n* = 43	*n* = 32	−25.6%	3.91 ± 2.3	2.91 ± 2.3	−1.03–3.03	0.316
PACI	*n* = 50	*n* = 55	+10.0%	Median: 4 (IQR 3.5)	Median: 4 (IQR 2.5)	N/A	0.616
LACI	*n* = 62	*n* = 59	−4.8%	5.64 ± 1.9	5.36 ± 1.8	−1.38–1.93	0.734
POCI	*n* = 42	*n* = 24	−42.9%	3.82 ± 1.7	2.18 ± 1.2	0.33–2.94	0.017
Infarct NOS	*n* = 7	*n* = 4	−42.9%	Median: 1 (IQR 1)	Median: 0 (IQR 1)	N/A	0.350
Thrombolysis	*n* = 45	*n* = 25	−44.4%	Median: 4 (IQR 1.5)	Median: 3 (IQR 2)	N/A	0.018

**Table 2 jcm-10-01357-t002:** Stroke Subtypes in the Pre-lockdown and Lockdown Cohorts. Proportion of stroke subtypes in the pre-lockdown and lockdown cohorts, with no significant differences identified. The proportion of ischaemic strokes thrombolysed was reduced in the lockdown cohort, but this was non-significant. *p* values were generated with Fisher’s Exact Test; values <0.05 were deemed statistically significant. ICH: intracerebral haemorrhage, TACI: total anterior circulation infarct, PACI: partial anterior circulation infarct, LACI: lacunar infarct, POCI: posterior circulation infarct: NOS: not otherwise specified.

Diagnosis	Pre-Lockdown		Lockdown		Change	*p* Value
Ischaemic Stroke	*n* = 204/223	(91.5%)	*n* = 174/200	(87.0%)	−4.5%	0.822
ICH	*n* = 19/223	(8.5%)	*n* = 26/200	(13.0%)	+4.5%	0.822
TACI	*n* = 43/204	(21.1%)	*n* = 32/174	(18.4%)	−2.7%	0.521
PACI	*n* = 50/204	(24.5%)	*n* = 55/174	(31.6%)	+7.1%	0.135
LACI	*n* = 62/204	(30.4%)	*n* = 59/174	(33.9%)	+3.5%	0.507
POCI	*n* = 42/204	(20.5%)	*n* = 24/174	(13.8%)	−6.7%	0.103
Infarct NOS	*n* = 7/204	(3.4%)	*n* = 4/174	(2.3%)	−1.1%	0.557
Ischaemic Strokes Thrombolysed	*n* = 45/204	(22.1%)	*n* = 25/174	(14.4%)	−7.7%	0.063

**Table 3 jcm-10-01357-t003:** Stroke Onset-to-Presentation Time in the Pre-lockdown and Lockdown Cohorts. Median time (in hours) from stroke symptom onset to hospital presentation between the pre-lockdown and lockdown cohorts, with no significant differences identified. *p* values were generated with Wilcoxon’s signed-rank Test; values <0.05 were deemed statistically significant. ICH: intracerebral haemorrhage, TACI: total anterior circulation infarct, PACI: partial anterior circulation infarct, LACI: lacunar infarct, POCI: posterior circulation infarct: NOS: not otherwise specified, h: hours, IQR: interquartile range, NA: not available.

	Pre-Lockdown Onset-To-Presentation Time			Lockdown Onset-To-Presentation Time				
Diagnosis	Size	Median (h)	IQR (h)	Size	Median (h)	IQR (h)	Change (h)	*p* Value
Stroke	*n* = 175	4.83	13.1	*n* = 165	5.42	13.1	+0.59	0.323
Ischaemic Stroke	*n* = 161	5.22	13.2	*n* = 142	6.11	13.0	+0.89	0.249
ICH	*n* = 14	2.19	7.5	*n* = 23	2.68	12.0	+0.49	0.673
TACI	*n* = 35	2.25	7.0	*n* = 26	2.88	5.5	+0.63	0.218
PACI	*n* = 39	3.68	8.7	*n* = 40	4.28	11.6	+0.60	0.222
LACI	*n* = 51	8.12	14.6	*n* = 53	6.98	14	−1.14	0.833
POCI	*n* = 31	11.82	15.6	*n* = 20	13.67	42.5	+1.85	0.401
Infarct NOS	*n* = 5	2.25	0.91	*n* = 2	8.04	3.89	+5.79	N/A

**Table 4 jcm-10-01357-t004:** Stroke Mimic Admissions and TIA Referrals in the Pre-lockdown and Lockdown Cohorts. Stroke mimic/non-stroke admissions as well as TIA referrals in the pre-lockdown and lockdown cohorts. A highly significant reduction in weekly stroke mimic admissions was noted in the lockdown cohort. There were significantly fewer weekly referrals made in the lockdown cohort; the number of referrals diagnosed as TIA remained similar, with a significant reduction in referrals deemed to be non-TIA. *p* values were generated with Student’s t-test if data were normally distributed, otherwise Wilcoxon’s signed-rank was performed; values <0.05 were deemed statistically significant. TIA: transient ischaemic attack, SD: standard deviation, IQR: interquartile range, CI: confidence intervals, NA: not available.

	11-Week Total			Weekly Admissions			
Diagnosis	Pre-Lockdown	Lockdown	Change	Pre-Lockdown (Mean ± SD)	Lockdown (Mean ± SD)	95% CI	*p* Value
Stroke-Mimic Admissions	*n* = 193	*n* = 102	−47.2%	17.5 ± 3.3	9.3 ± 4.1	4.95–11.6	<0.001
Total Referrals	*n* = 152	*n* = 105	−30.9%	Median: 12 (IQR 6.5)	Median: 9 (IQR 5.5)	N/A	0.037
TIA Referrals	*n* = 57	*n* = 53	−7.02%	Median: 4 (IQR 2.5)	Median: 5 (IQR 4.5)	N/A	0.552
Non-TIA Referrals	*n* = 95	*n* = 52	−54.7%	8.64 ± 3.1	4.73 ± 2.1	4.73–8.64	0.002

## Data Availability

Supporting data is available on correspondence.

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
