# Peer review of "The Effect of the 2019 Novel Coronavirus Pandemic on Stroke and TIA Patient Admissions: Perspectives and Risk Factors"

_jcm, 2021, doi:10.3390/jcm10071357_

Round 1
Reviewer 1 Report
Dear Editor,
Dear Author,
I did like to read the present paper.
It addresses the decline in presentation for stroke during the Covod 19 pandemic in hospital in UK.
Unfortunately, it confirm the trend described in many countries. Less medical consultations caused by the pandemic. The colleagues from the UK present important data, the data are well presented, the graphs are well designed, I like them.
In my opinion, the manuscript should be accepted for publication. I have only some minor revision to suggest:
-
- In the abstract: in the results the authors report on stroke mimics, but no operational definition for that is mentioned in the methods. Similar this is the case in the main manuscript.
- Abbreviations in tables should be readable independently, so they must be explained entirely in the legend.
- The introduction part of the discussion contains an enumeration of results from other studies. Why that? Please begin the discussion emphasising YOUR results
- Line 233: please delete “This study” and change it into “The presented study” or “Current study” or “Our study”.
Author Response
Reviewer one
Dear Editor,
Dear Author,
I did like to read the present paper.
It addresses the decline in presentation for stroke during the Covod 19 pandemic in hospital in UK.
Unfortunately, it confirm the trend described in many countries. Less medical consultations caused by the pandemic. The colleagues from the UK present important data, the data are well presented, the graphs are well designed, I like them.
In my opinion, the manuscript should be accepted for publication. I have only some minor revision to suggest:
-
- In the abstract: in the results the authors report on stroke mimics, but no operational definition for that is mentioned in the methods. Similar this is the case in the main manuscript.
- Abbreviations in tables should be readable independently, so they must be explained entirely in the legend.
- The introduction part of the discussion contains an enumeration of results from other studies. Why that? Please begin the discussion emphasising YOUR results
- Line 233: please delete “This study” and change it into “The presented study” or “Current study” or “Our study”.
Many thanks for your kind comments on the paper, we have actioned your comments as below:
- Thank you, this has been addressed in the abstract and in the main text lines 97-98.
- Many thanks for this, all abbreviations have been removed or defined for our tables, amendments have been made in Table 1, and also in Figures 1 and 3.
- We had included this to set our paper in context of the wider literature from larger populations. We have changed the opening to focus more on our results, many thanks for this.
- This has been amended, many thanks.
Reviewer 2 Report
As described, this is a single-center experience in a non-hub center. Referring to the fall in the posteriori ischaemias, it would be interesting to know if in the referring center a specular increase of the basilar tip occlusion is observed. If so, it could be explained as a changing in the pre-hospital evaluation of the most severe clinical presentation patients (as basilar-tip occlusions).
In the evaluation of the italian incidence of stroke in COVID paitents, it is useful to cite:
- Frisullo G, Brunetti V, Di Iorio R, Broccolini A, Caliandro P, Monforte M, Morosetti R, Piano C, Pilato F, Calabresi P, Della Marca G; STROKE TEAM Collaborators. Effect of lockdown on the management of ischemic stroke: an Italian experience from a COVID hospital. Neurol Sci. 2020 Sep;41(9):2309-2313. doi: 10.1007/s10072-020-04545-9. Epub 2020 Jul 6. PMID: 32632635; PMCID: PMC7338130.
- Asteggiano F, Divenuto I, Ajello D, Gennaro N, Santonocito O, Marcheselli S, Balzarini L, Nuzzi NP, Politi LS. Stroke management during the COVID-19 outbreak: challenges and results of a hub-center in Lombardy, Italy. Neuroradiology. 2021 Jan 7:1–5. doi: 10.1007/s00234-020-02617-3. Epub ahead of print. PMID: 33410950; PMCID: PMC7788174.
Author Response
Reviewer two
As described, this is a single-center experience in a non-hub center. Referring to the fall in the posteriori ischaemias, it would be interesting to know if in the referring center a specular increase of the basilar tip occlusion is observed. If so, it could be explained as a changing in the pre-hospital evaluation of the most severe clinical presentation patients (as basilar-tip occlusions).
In the evaluation of the italian incidence of stroke in COVID paitents, it is useful to cite:
- Frisullo G, Brunetti V, Di Iorio R, Broccolini A, Caliandro P, Monforte M, Morosetti R, Piano C, Pilato F, Calabresi P, Della Marca G; STROKE TEAM Collaborators. Effect of lockdown on the management of ischemic stroke: an Italian experience from a COVID hospital. Neurol Sci. 2020 Sep;41(9):2309-2313. doi: 10.1007/s10072-020-04545-9. Epub 2020 Jul 6. PMID: 32632635; PMCID: PMC7338130.
- Asteggiano F, Divenuto I, Ajello D, Gennaro N, Santonocito O, Marcheselli S, Balzarini L, Nuzzi NP, Politi LS. Stroke management during the COVID-19 outbreak: challenges and results of a hub-center in Lombardy, Italy. Neuroradiology. 2021 Jan 7:1–5. doi: 10.1007/s00234-020-02617-3. Epub ahead of print. PMID: 33410950; PMCID: PMC7788174.
Many thanks for your kind comments.
Unfortunately for our studied cohort we did not have the data available for basilar tip occlusions. However, this is an interesting question, and we will aim to collect this data to form a future research project.
Thank you for mentioning these interesting papers. We have incorporated the first paper in lines 238-239. We also enjoyed reading the second paper which provides a useful insight; however, we have not included this in our paper as it describes restructuring of stroke services during COVID-19, offering a different focus to our work.
Reviewer 3 Report
Here are the main comments to the manuscript:
1. The term "2019-nCov" is not precise, instead of this the term "COVID-19" is used,
2. Abbreviations should not be used in the abstract, even if they are commonly known,
3. L40 - the authors rightly state that COVID-19 is associated with hypercoagulability, however, as recent studies show, routinely used laboratory parameters are not sufficient to assess this phenomenon in COVID-19. The best here are TEG and TEM, hence the need to add a new literature item to this sentence (Thromb Haemost DOI: 10.1055/a-1346-3178).
4. Whether intravenous thrombolysis is performed in your hospital around the clock. The question arises from the fact that mechanical thrombectomy is performed over an interval of hours. Generally, it is a surprise for me, we perform this procedure 24 hours a day in my hospital.
5. Whether the authors used the patient inclusion/exclusion criteria.
6. The authors should describe in detail what the lockdown looked like and what restrictions were introduced there. Without this data, the analysis of the results is not relevant.
7. In the text of the manuscript there is no information about the consent of the bioethics committee.
8. Why the authors presented the Oxfordshire Community Stroke Project classification and did not present other data such as age, gender, NIHSS, mRS, BI, TOAST.
9. In table 1 - what is included in the term "stroke"?
10. Table 2 - what is the purpose of its presentation and what does it bring to the results?
11. It is difficult to understand the results if the authors have not previously defined ischemic stroke, hemorrhagic stroke, TIA and other clinically significant patient conditions.
12. The results of the questionnaires are not presented carefully.
13. Literature is very poor.
Author Response
Reviewer three
Here are the main comments to the manuscript:
1. The term "2019-nCov" is not precise, instead of this the term "COVID-19" is used,
2. Abbreviations should not be used in the abstract, even if they are commonly known,
3. L40 - the authors rightly state that COVID-19 is associated with hypercoagulability, however, as recent studies show, routinely used laboratory parameters are not sufficient to assess this phenomenon in COVID-19. The best here are TEG and TEM, hence the need to add a new literature item to this sentence (Thromb Haemost DOI: 10.1055/a-1346-3178).
4. Whether intravenous thrombolysis is performed in your hospital around the clock. The question arises from the fact that mechanical thrombectomy is performed over an interval of hours. Generally, it is a surprise for me, we perform this procedure 24 hours a day in my hospital.
5. Whether the authors used the patient inclusion/exclusion criteria.
6. The authors should describe in detail what the lockdown looked like and what restrictions were introduced there. Without this data, the analysis of the results is not relevant.
7. In the text of the manuscript there is no information about the consent of the bioethics committee.
8. Why the authors presented the Oxfordshire Community Stroke Project classification and did not present other data such as age, gender, NIHSS, mRS, BI, TOAST.
9. In table 1 - what is included in the term "stroke"?
10. Table 2 - what is the purpose of its presentation and what does it bring to the results?
11. It is difficult to understand the results if the authors have not previously defined ischemic stroke, hemorrhagic stroke, TIA and other clinically significant patient conditions.
12. The results of the questionnaires are not presented carefully.
13. Literature is very poor.
Thank you for reviewing the paper and for your comments. We have addressed these as below:
- We have changed this throughout the paper, many thanks.
- Many thanks, this has been changed.
- This reference has been added, many thanks.
- We offer 24/7 IV thrombolysis, and a 7 day thrombectomy service.
- For the admission data analysis all admitted patients were included in the analysis, I have added this in the methods lines 90-91, many thanks. Our inclusion criteria for our patient questionnaire are stated in lines 111-115.
- Many thanks, this has been added to lines 50-52.
- The admissions data analysis was performed on data already collected to populate Safe Implementation of Treatment in Stroke (SITS) where we were identified as an exemplar centre and Sentinel Stroke National Audit Programme (SSNAP) registries, so their collection is under the ethical approval of those studies in 2019. Our patient questionnaire was implemented as an urgent clinical tool to routine stroke clerkings to allow for personalised patient advice on discharge on ACT-FAST advice in a radically different landscape. The results of this were then subsequently investigated and are reported here. This investigation of the patient questionnaires was registered with the trust as a clinical audit.
- Thank you for raising this. The Bamford classification was used as our outcome measure as it allows the most useful indicator of severity and future prognosis following these strokes. Demographics of those who suffered strokes in this timeframe were not investigated, however this is an interesting area for further work.
- This combined all ischaemic strokes and ICH, I have clarified this in the table, many thanks.
- Although results are non-significant, we felt their lack of significance was interesting and important to report. Table 2 was included to highlight important negative findings and provides a more digestible interface for the reader.
- We feel it is reasonable to assume basic knowledge of these conditions in the readers of your journal. However, if this is not acceptable, we are happy to add definitions of these.
- This was not a point raised by the other reviewers; however, we would be happy to include a further table to present the results of the questionnaire if this would be beneficial.
- We have updated our references on kind additions suggested by the reviewers, many thanks for your contributions.
Round 2
Reviewer 3 Report
The authors have addressed all the comments of the reviewer and revised the manuscript accordingly.